# Towards Permutation Invariant Learning with High-Dimensional Particle Filters

## Abstract

Sequential learning in deep models often suffers from challenges such as catastrophic forgetting and loss of plasticity. This effect is largely due to the permutation dependence of gradient-based algorithms, where the order of training data affects the learning outcome. In this work, we introduce a novel learning framework based on high-dimensional particle filters that yields approximately permutation-invariant results. We theoretically demonstrate that particle filters are approximately invariant to the sequential ordering of training minibatches or tasks, offering a principled solution to mitigate catastrophic forgetting and loss-of-plasticity. Next, we develop an efficient particle filter for optimizing high-dimensional models, combining the strengths of Bayesian methods with gradient-based optimization. Finally, through extensive experiments on continual supervised and reinforcement learning benchmarks, including SplitMNIST, SplitCIFAR100, and ProcGen, we empirically demonstrate that our method consistently improves performance, while reducing variance compared to standard baselines.

## 1 Introduction

**What is the optimal order for training data?** This question is fundamental to understanding how the sequencing of training data impacts machine learning model performance. In sequential learning settings, such as continual learning and lifelong learning, the sequencing of training data plays a crucial role in determining model performance. When models are trained on ordered minibatches of data, poor orderings can result in catastrophic forgetting and loss of plasticity (Wang et al., 2024; Abel et al., 2023).

In continual learning, models process tasks in a specific sequence. Unlike conventional training, where minibatch data is randomized, continual learning often relies on a strict sequence, making models prone to overfitting on newer tasks while losing performance on older tasks. This is known as *catastrophic forgetting*, where new information erases prior knowledge, severely degrading performance on earlier tasks (Kim & Han, 2023; van de Ven et al., 2022).

Similarly, in lifelong reinforcement learning (LRL), agents must adapt to new tasks sequentially. The order in which these tasks are presented can lead to *loss of plasticity*, limiting the agent's ability to adapt to new environments (Muppidi et al., 2024; Lyle et al., 2022; Abbas et al., 2023; Sokar et al., 2023). This poor ordering can further manifest as *negative transfer* or *primacy bias*, where learning earlier tasks biases the agent towards those tasks, impeding adaptation to new tasks (Nikishin et al., 2022; Ahn et al., 2024).

To address these challenges, we propose a shift in perspective, viewing the problem through the lens of *permutation invariance*. By developing learning algorithms that are invariant to the order of data presentation, we can mitigate catastrophic forgetting and loss of plasticity. Our key insight is the use of *particle filters*, a probabilistic tool widely used in state estimation, to achieve this goal. Particle filters excel at dynamically estimating system states from noisy data and are grounded in Bayesian inference (Thrun et al., 2005; Doucet et al., 2001b; Jonschkowski et al., 2018; Karkus et al., 2021; Corenflos et al., 2021; Pulido & van Leeuwen, 2019; Maken et al., 2022; Boopathy et al., 2024). However, their application to modern machine learning has been limited due to scalability issues in high-dimensional settings. In contrast, gradient-based optimization techniques such as gradient descent efficiently handle high-dimensional spaces but lack the probabilistic framework offered by particle filters.

In this work, we bridge this gap by proposing a novel particle filter designed specifically for high-dimensional learning. We show that by adapting particle filters to high-dimensional learning problems, we can achieve more robust, approximately permutation-invariant learning. Our approach provides a new perspective on training in sequential settings and also addresses the core challenges of catastrophic forgetting and loss of plasticity in a principled manner. Our contributions are three-fold:

- Theoretically, we demonstrate that particle filters enable approximate permutation-invariant learning, where the algorithm's output remains nearly the same regardless of the training data order. We further show that this property naturally mitigates catastrophic forgetting and loss of plasticity.

- We introduce a simple, gradient-based particle filter specifically tailored for high-dimensional parameter spaces. This filter retains the essential features of traditional particle filters while being computationally efficient and well-suited for typical machine-learning optimization tasks.

- Through empirical evaluations on continual learning and lifelong reinforcement learning benchmarks, including SplitMNIST, SplitCIFAR100, and ProcGen, we show that our proposed particle filter achieves better performance and reduced variance over permutations compared to standard baselines. Additionally, we demonstrate that integrating this particle filter with continual learning and LRL methods increases overall performance and reduces performance variance.

## 2 RELATED WORK

### 2.1 PARTICLE FILTERS

Particle filters, or sequential Monte Carlo methods, are widely used for state estimation in non-linear and non-Gaussian settings. They represent probability distributions through a set of samples (particles), providing flexibility in capturing complex dynamics (Doucet et al., 2001a). In fields such as robotics, particle filters have been applied successfully to localization and mapping problems, where they handle uncertainty and non-linearities effectively (Thrun, 2002). However, a key limitation is their scalability: as the dimensionality of the problem increases, the number of particles needed grows exponentially, making them less practical in high-dimensional spaces like those in machine learning (Bengtsson et al., 2008). Recent efforts have focused on improving particle filter scalability through adaptive resampling and dimensionality reduction techniques (Li et al., 2015), but these approaches have not fully bridged the gap for large-scale machine learning applications. Our work addresses this gap by proposing a high-dimensional particle filter that is computationally efficient and well-suited for machine learning tasks.

### 2.2 BAYESIAN MODEL AVERAGING

Bayesian model averaging (BMA) is a powerful technique for integrating uncertainty into model predictions by averaging across multiple models (Hoeting et al., 1999; Wasserman, 2000). By weighting model predictions based on their posterior probabilities, BMA can provide more robust predictions and better capture model uncertainty compared to single-model approaches (Raftery et al., 2005). In modern machine learning, BMA has been employed to enhance performance and uncertainty estimation, notably in ensemble techniques (Lakshminarayanan et al., 2017; Wortsman et al., 2022). In particular, some works consider ensembling in the context of continual learning (Rypeść et al., 2024; Carta et al., 2023; Wen et al., 2020), and more generally, Bayesian methods are known to avoid catastrophic forgetting (Ritter et al., 2018; Li et al., 2020; Kochurov et al., 2018; Farquhar & Gal, 2019). In this work, we apply the benefits of Bayesian methods to the more general problem of permutation-invariant learning. We describe a particular particle filter that functions as a BMA technique, and demonstrate its applicability to permutation-invariant learning theoretically and empirically.

## 3 PARTICLE FILTERS FOR LEARNING PROBLEMS

In this section, we first theoretically demonstrate two beneficial properties of particle filters generally on learning problems, namely 1) approximate permutation-invariance and 2) avoidance of catastrophic forgetting and loss of plasticity. We then describe a particular particle filter suitable for high-dimensional learning problems.

### 3.1 SETUP

Consider a learning problem that provides a sequence of loss functions of model parameters, and the goal of learning is to minimize the sum of the loss functions. For instance, the loss function at each time step might correspond to the cross-entropy loss on a minibatch of points for a classification problem. We denote the model parameters at time $t$ as $x_t \in \mathbb{R}^d$ and the loss function at time $t$ as $L_t \in \mathbb{R}^d \to \mathbb{R}$ (a mapping from $d$ dimensions to 1 dimension). The goal is to find an $x$ minimizing $\sum_{t=1}^{T} L_t(x)$, where $T$ is the total number of updates.

How can we apply particle filters to this learning problem? To do this, we suppose that instead of learning a single model, we learn a *distribution* of models following a Bayesian approach. Specifically, suppose that at time $t = 0$, we initially start with a prior distribution of candidate models; each $L_t$ corresponds to an observation that updates the likelihood of each model. Specifically, we suppose that the likelihood of the model $x$ is set as:

$$P(L_t|x) = e^{-L_t(x)} \tag{1}$$

This likelihood function increases the likelihood of models that achieve lower loss values. We denote the prior distribution of models as $p_0$ and the posterior distribution after having observed $L_1$ through $L_t$ as $p_t$. Then, $p_T$ is given by:

$$p_T(x) = Z_T p_0(x) \Pi_{t=1}^{T} P(L_t|x) = Z_T p_0(x) e^{-\sum_{t=1}^{T} L_t(x)} \tag{2}$$

where $Z_T$ is a normalization factor that ensures $p_T$ integrates to 1. Observe that this posterior places high density in regions where the summed loss is low. Particle filters enable the computation of $p_T(x)$ by incrementally computing estimates of $p_t(x)$. Specifically, given $p_t(x)$, we may compute $p_{t+1}(x)$ as:

$$p_{t+1}(x) = \frac{p_t(x)P(L_{t+1}|x)}{\int p_t(x')P(L_{t+1}|x')dx'} = \frac{p_t(x)e^{-L_{t+1}(x)}}{\int p_t(x')e^{-L_{t+1}(x')}dx'} \tag{3}$$

This Bayesian update equation may often be intractable to compute exactly, particularly when $p_t$ does not have a known parametric form. Instead of tracking $p_t$ exactly, particle filters track an estimate $\hat{p}_t$ instead:

$$\hat{p}_t(x) = \sum_i w_t^{(i)} \delta(x - x_t^{(i)}) \tag{4}$$

where $\delta$ is a delta function, $x_t^{(i)}$ represents the $i$th particle at time $t$ and $w_t^{(i)}$ represents the weight of the particle at time $t$. Each particle filter then has a different method of estimating the Bayesian update of Equation 3. After all updates are complete, an ensemble of particles is available, each of which is an estimate of the global minimizer of $\sum_{t=1}^{T} L_t(x)$. We denote the output distribution of a particle filter initialized at $\hat{p}_0$ trained on a sequence of loss functions $L_1, ...L_T$ as $\hat{p}_0[L_1, ...L_T]$. Since particle filters aim to approximate Bayesian updates, we suppose that each update outputs a set of particles close to the true posterior. To formalize this, suppose that there exists a symmetric, non-negative discrepancy measure $D(p, q)$ that satisfies the triangle inequality:

$$D(p, q) \leq D(p, r) + D(r, q) \tag{5}$$

for all $p, q, r$. Furthermore, suppose $D(p, p) = 0$ for all $p$. Now, suppose that the particle filter satisfies the following two conditions:

$$D(\hat{p}[L], \hat{q}[L]) \leq CD(\hat{p}, \hat{q}) \tag{6}$$

and

$$D(\hat{p}[L], \frac{p(\cdot)e^{-L(\cdot)}}{\int p(x)e^{-L(x)}dx}) \leq CD(\hat{p}, p) + \epsilon \tag{7}$$

for some constants $C$ and $\epsilon$ where $\hat{p}[L]$ denotes $\hat{p}$ after being updated on $L$. $C$ can be interpreted as the stability of the particle filter update, while $\epsilon$ can be viewed as error between each particle filter update and the true Bayesian update. This allows the discrepancy at time $T$ to be bounded as:

$$D(\hat{p}_0[L_1, ... L_T], \frac{p(\cdot)e^{-\sum_t L_t(\cdot)}}{\int p(x)e^{-\sum_t L_t(x)}dx}) \le C^T D(\hat{p}_0, p_0) + \epsilon \frac{C^T - 1}{C - 1} \tag{8}$$

This decomposes the discrepancy at time $T$ as a term depending on the initial discrepancy $D(\hat{p}_0, p_0)$ and a term depending on the incremental discrepancy $\epsilon$.

## 3.2 PERMUTATION-INVARIANCE

Next, we demonstrate that particle filters are approximately *permutation-invariant*: they produce an output that is nearly invariant to the ordering of loss functions $L_t$. We show that $\hat{p}_0[L_1, ... L_T]$ is similar to $\hat{p}_0[L_{\sigma_1}, ... L_{\sigma_T}]$ where $\sigma_1, \sigma_2, ... \sigma_T$ is a permutation of $1, 2, ... T$.

**Theorem 1.** *Suppose $\sigma_1, \sigma_2, ... \sigma_T$ is a permutation of $1, 2, ... T$ such that $N$ swaps of adjacent elements are required to convert $\sigma_1, \sigma_2, ... \sigma_T$ to $1, 2, ... T$. Denote the initialized particle filter as $\hat{p}_0$. Then,*

$$D(\hat{p}_0[L_1, ... L_T], \hat{p}_0[L_{\sigma_1}, ... L_{\sigma_T}]) \le 2N\epsilon C^{T-2}(C + 1) \tag{9}$$

See Appendix A for a proof. This result demonstrates that, when $C \approx 1$ and $\epsilon$ is sufficiently small, particle filters are approximately permutation invariant. Note that the size of the bound grows exponentially with $T$; thus, permutation invariance can rapidly collapse when $C >> 1$. However, recall that $C$ represents the stability of the particle filter update (i.e. how much small variations in particles propagate through updates). If $C \approx 1$, then small fluctuations in the initialization of particles do not significantly affect the outcome after updating. We expect this to be a reasonable assumption for many practical particle filters. Standard learning algorithms such as gradient descent are notably *not* permutation-invariant: they tend to be highly dependent on the ordering of data points. Permutation-invariance enables learning algorithms with less stochastic outputs: in a perfectly permutation-invariant particle filter, the only potential sources of randomness are the initial selection of particles and the randomness in the particle filter updates themselves.

## 3.3 AVOIDING CATASTROPHIC FORGETTING AND LOSS OF PLASTICITY

Now, we demonstrate that particle filters *naturally* avoid catastrophic forgetting and loss of plasticity. Catastrophic forgetting can be formalized in our framework as the phenomenon where a learning algorithm trained on a sequence of losses $L_1, ... L_T$ performs poorly on the earlier losses it is trained on. Similarly, loss of plasticity corresponds to performing poorly on later losses. We provide an upper bound on the loss at any point in training:

**Theorem 2.** *Suppose that all loss functions are bounded in range $[0, \infty)$. Suppose that there exists a constant $k$ such that for all loss functions $L$ and distributions $p, q$:*

$$\mathbb{E}_{x \sim p}[L(x)] - \mathbb{E}_{x \sim q}[L(x)] \le kD(p, q) \tag{10}$$

*Then,*

$$\mathbb{E}_{x \sim \hat{p}_0[L_1, L_2, ... L_T]}[L_i(x)] \le \beta \mathbb{E}_{x \sim \hat{p}_0[L_1, ... L_{i-1}, L_{i+1}, ... L_T]}[L_i(x)] + 2kT\epsilon C^{T-2}(C + 1) \tag{11}$$

*where $\beta$ is a constant satisfying:*

$$\beta \ge \frac{\mathbb{E}_{x \sim \hat{p}}[L(x)]}{\mathbb{E}_{x \sim \hat{p}[L]}[L(x)]} \tag{12}$$

*for all $L$ and $\hat{p}$.*

See Appendix B for a proof. We make two key assumptions in this theorem: the difference in average loss under two different distributions can be bounded by $D$ and that each step in the particle filter reduces the loss on which it is trained by at least a fixed factor. We believe that the first assumption may in many cases be reasonable if the loss function is sufficiently slow-changing: small changes in the distribution over $x$ should not change the average loss value. The second assumption may also be reasonable under many settings for effective particle filters as well as other standard learning algorithms; with a fixed loss function, it corresponds to a linear convergence rate. Gradient descent, for example, satisfies this assumption on loss functions satisfying the Polyak-Łojasiewicz inequality. The resulting bound on the loss $L_i$ guarantees that the performance is guaranteed to improve by a fixed factor relative to not observing $L_i$ plus an additional error term.

### 3.4 GRADIENT-BASED PARTICLE FILTER

Given the desirable properties of particle filters in learning problems, here, we describe a particular particle filter well suited to the high-dimensional spaces found in most machine learning settings. We derive this particle filter by taking the following steps: we assume that our particle filter represents a mixture of Gaussians with each Gaussian centered at a particle. We then show that when the Gaussian covariances are sufficiently small, the optimal Bayesian update on the mixture of Gaussians results in (approximately) another mixture of Gaussians with updated means and mixture weights. Thus, we are able to represent the optimal Bayesian update by simply updating the particle positions and weights.

---

**Algorithm 1** Gradient-based particle filter

**Input:** Initial particles $\{x_0^{(i)}\}_{i=1}^N$, Loss functions $L_1, ... L_T$, Variance $\sigma^2$
**Output:** Updated particles $\{x_T^{(i)}, w_T^{(i)}\}_{i=1}^N$
**for** $t = 0$ to $T - 1$ **do**
    $w_0^{(i)} = 1$
    **for** $i = 1$ to $N$ **do**
        $x_{t+1}^{(i)} = x_t^{(i)} - \sigma^2 \nabla L_{t+1}(x_t^{(i)})$
        $w_{t+1}^{(i)} = w_t^{(i)} e^{-\frac{L_{t+1}(x_{t+1}^{(i)}) + L_{t+1}(x_t^{(i)})}{2}}$
    **end for**
**end for**
$S = 0$
**for** $i = 1$ to $N$ **do**
    $S = S + w_T^{(i)}$
**end for**
**for** $i = 1$ to $N$ **do**
    $w_T^{(i)} = w_T^{(i)}/S$
**end for**

---

Now we walk through the steps: suppose that our particle filter's particle distribution $\hat{p}_t(x) = \sum_i w_t^{(i)} \delta(x - x_t^{(i)})$ represents another distribution $\tilde{p}_t(x)$ constructed as:

$$\tilde{p}_t(x) = Z \sum_i w_t^{(i)} e^{-\frac{||x - x_t^{(i)}||^2}{2\sigma^2}} \tag{13}$$

where $\sigma$ is a variance parameter and $Z$ is a normalizing constant. Essentially, $\tilde{p}_t(x)$ replaces each delta function in $\hat{p}_t(x)$ with a isotropic Gaussian. We derive the particle filter's update on $\hat{p}_t(x)$ as an approximation of the optimal Bayesian update applied to $\tilde{p}_t(x)$. Observe that given a prior of $\tilde{p}_t(x)$ and the loss function $L_{t+1}$, the posterior is proportional to:

$$e^{-L_{t+1}(x)} \sum_i w_t^{(i)} e^{-\frac{||x - x_t^{(i)}||^2}{2\sigma^2}} \tag{14}$$

We manipulate this expression until it can be expressed in the form of Equation 13. First, we make a linear approximation of $L_{t+1}$ centered at each particle $x_t^{(i)}$:

$$L_{t+1}(x) \approx L_{t+1}(x_t^{(i)}) + \nabla L_{t+1}(x_t^{(i)})^T (x - x_t^{(i)}) \tag{15}$$

Note that the approximation error is on the order of $\sigma^2$. Pulling $e^{-L_{t+1}(x)}$ into the summation and applying this approximation:

$$\sum_i w_t^{(i)} e^{-\frac{||x - x_t^{(i)}||^2}{2\sigma^2} - L_{t+1}(x_t^{(i)}) - \nabla L_{t+1}(x_t^{(i)})^T (x - x_t^{(i)})} \tag{16}$$

Grouping terms and completing the square in the exponent:

$$\sum_i w_t^{(i)} e^{-\frac{||x - x_{t+1}^{(i)}||^2}{2\sigma^2}} e^{\frac{||x_{t+1}^{(i)}||^2 - ||x_t^{(i)}||^2}{2\sigma^2} - L_{t+1}(x_t^{(i)}) + \nabla L_{t+1}(x_t^{(i)})^T x_t^{(i)}} \tag{17}$$

where $x_{t+1}^{(i)} = x_t^{(i)} - \sigma^2 \nabla L_{t+1}(x_t^{(i)})$. Simplifying the constant terms:

$$\sum_i w_t^{(i)} e^{-\frac{||x-x_{t+1}^{(i)}||^2}{2\sigma^2}} e^{\frac{\sigma^2||\nabla L_{t+1}(x_t^{(i)})||^2}{2} - L_{t+1}(x_t^{(i)})} \tag{18}$$

Observe that under our linear approximation, $L_{t+1}(x_{t+1}^{(i)}) \approx L_{t+1}(x_t^{(i)}) - \sigma^2 ||\nabla L_{t+1}(x_t^{(i)})||^2$ (with approximation error on the order of $\sigma^4$). Thus, we may write the expression as:

$$\sum_i w_t^{(i)} e^{-\frac{||x-x_{t+1}^{(i)}||^2}{2\sigma^2}} e^{-\frac{L_{t+1}(x_{t+1}^{(i)})+L_{t+1}(x_t^{(i)})}{2}} \tag{19}$$

Finally, we define $w_{t+1}^{(i)} = w_t^{(i)} e^{-\frac{L_{t+1}(x_{t+1}^{(i)})+L_{t+1}(x_t^{(i)})}{2}}$ to arrive at our final approximation of the posterior:

$$\sum_i w_{t+1}^{(i)} e^{-\frac{||x-x_{t+1}^{(i)}||^2}{2\sigma^2}} \tag{20}$$

Due to our approximations, we may expect a proportional error up to roughly $e^{\sigma^2}$, which approaches 1 as $\sigma^2 \to 0$. We represent this posterior with particles $x_{t+1}^{(i)}$ and respective weights $w_{t+1}^{(i)}$. We summarize the update equations of this particle filter below:

$$x_{t+1}^{(i)} = x_t^{(i)} - \sigma^2 \nabla L_{t+1}(x_t^{(i)}) \tag{21}$$

$$w_{t+1}^{(i)} = w_t^{(i)} e^{-\frac{L_{t+1}(x_{t+1}^{(i)})+L_{t+1}(x_t^{(i)})}{2}} \tag{22}$$

Algorithm 1 shows the full pseudocode of the filter. For efficiency, we do not normalize the weights of the particles at each iteration; this can be done once at the end of training. Intuitively, this filter updates the positions of the particles with gradient descent, but reweights the particles based on their performance at the old and new points, with lower-loss particles weighted higher. We highlight that the particle filter operates *independently* on all points except for the final step of weight normalization. Practically, this means the filter can be run in parallel, making the computation time independent of the number of particles. Like gradient descent and other gradient-based optimization procedures, this particle filter is well suited for optimization on high-dimensional spaces, while retaining the properties of particle filters outlined in the prior sections such as approximate permutation-invariance and avoidance of catastrophic forgetting. Appendix E Figure 2 illustrates how our method converges to well-performing regions of the parameter space over time.

**Theoretical guarantees** Observe that this particle filter is built on gradient descent; thus it inherits the theoretical convergence guarantees of gradient descent. In particular, unlike gradient-free particle filters, this approach is suited for high-dimensional spaces just as gradient-based optimization methods require many fewer iterations relative to gradient-free methods in high-dimensions. What separates this particle filter from simply a model average of $N$ models independently trained with gradient descent? Unlike a simple model average, this approach retains the Bayesian estimation properties of a traditional particle filter; namely, its output is an approximation of the Bayesian posterior. This allows it to maintain the desirable properties of particle filters described earlier. In particular, we can verify that it satisfies the assumptions of Theorem 1: namely, Equations 6 and 7. Equation 6 holds (for some constant $C$) when the particle filter update is Lipschitz continuous with respect to the discrepancy measure $D$. For our particle filter, this is true as long as the $L_t$ have bounded first and second derivatives. Equation 7 holds when the particle filter approximates a Bayesian update, which is true for ours by design.

Next, we demonstrate theoretically that the particle filter indeed maintains fidelity to the Bayesian updates it is based on. Specifically, we show in a simplified setting that the given two particles with the same prior probability at initialization, the particle filter produces an output exactly matching the probability ratios of the true posterior at the final particle locations:

**Theorem 3.** *Suppose particles $x^{(i)}$ and $x^{(j)}$ are initialized with the same prior probability:*

$$p_0(x_0^{(i)}) = p_0(x_0^{(j)}) \tag{23}$$

*Furthermore, suppose that all loss functions are linear:*

$$L_t(x) = g_t^T x + b_t \tag{24}$$

*Then,*

$$\frac{p_T(x_T^{(i)})}{p_T(x_T^{(j)})} = \frac{w_T^{(i)}}{w_T^{(j)}} \tag{25}$$

See Appendix C for a proof. This theorem guarantees that the particle filter indeed maintains weights in proportion to the true posterior distribution ($p_T(x_T^{(i)}) \propto w_T^{(i)}$). This, in fact, implies that, in the limit of infinitely many particles, the posterior over $x$ will *equal* the density over particles $p_T(x) = \sum_i w^{(i)} \delta(x - x_t^{(i)})$ (Doucet et al., 2001a). Thus, our particle filter achieves the best of both worlds: it can perform optimization in high-dimensions while also approximating Bayes optimal solutions.

## 4 EXPERIMENTS AND RESULTS

In this section, we empirically validate the approximate permutation-invariance of our gradient-based weighted particle filter (hereafter referred to as the weighted particle filter or WPF) and demonstrate its effectiveness in mitigating catastrophic forgetting and loss of plasticity across continual and lifelong learning benchmarks.

**Continual Learning Experiments:** We evaluate our weighted particle filter on continual learning benchmarks SplitMNIST (LeCun & Cortes, 2010), SplitCIFAR100 (Krizhevsky, 2009), and Proc-Gen (Cobbe et al., 2020). SplitMNIST is divided into 5 "super class" splits, and SplitCIFAR100 into 20, both for class-incremental learning. For ProcGen, we use image-action trajectory datasets from the games Starpilot, Fruitbot, and Dodgeball, partitioned into 15 levels sampled using the hard distribution shift mode. We compare our weighted particle filter against established continual learning methods: Synaptic Intelligence (SI), Elastic Weight Consolidation (EWC), and Learning Without Forgetting (LWF) (Zenke et al., 2017; Kirkpatrick et al., 2016; Li & Hoiem, 2016). Since our particle filter is architecture-agnostic, we also combine it with SI, EWC, and LWF to evaluate their joint effectiveness. In the ProcGen continual learning experiments, we use a supervised behavior cloning policy as our base model and compare it against other baseline particle filters also using supervised behavior cloning. All methods are implemented with identical architectures and learning parameters to ensure a fair comparison.

After training, we measure the average accuracy/return of both the weighted particle filter and the baseline models across all splits/levels using 10 different shuffled permutations of epoch splits. To assess permutation invariance, we calculate the task-specific variance in accuracy/return across these 10 permutations. These experiments are designed to evaluate the particle filter's resistance to catastrophic forgetting. Our weighted particle filter uses 100 particles, with test accuracy evaluated as a weighted average across particles. We compare this approach with three additional particle filter baselines: (1) a standard particle filter, (2) a gradient-based particle filter without weighting, and (3) traditional gradient descent (single particle). This allows us to assess the impact of particle weighting and the benefits of the Bayesian framework. The standard particle filter serves as a benchmark to evaluate performance on high-dimensional problems, and it operates by resampling particles based on their training loss performance. The gradient-based particle filter without weighting (referred to as averaging particles) is included as a baseline to determine the effectiveness of particle weighting. Full implementation details can be found in the appendix.

**LRL Experiments:** In the lifelong reinforcement learning setting, we adopt the setup from Muppidi et al. (2024) and conduct experiments on the ProcGen games Starpilot, Fruitbot, and Dodgeball. Distribution shifts are introduced by sampling new procedurally generated levels every 2 million time steps. The agent's performance is evaluated based on the average normalized return over the course of the lifelong experiment. Additionally, we measure the normalized variance across each level for 10 different permutations of the lifelong level sequences. We evaluate proximal policy optimization (PPO) (Schulman et al., 2017), along with other LRL methods designed to prevent loss of plasticity - specifically, PPO combined with TRAC (Muppidi et al., 2024) and PPO combined with EWC (Kessler et al., 2022)—each tested both with and without our weighted particle filter.

Table 1: Average accuracy and normalized variance across classes over 10 permutations for the weighted particle filter, standard particle filter methods, continual learning baselines, and continual learning baselines combined with the weighted particle filter on SplitMINST and SplitCIFAR100. Our methods are in *italics*. Best results in **bold**.

| Method | Avg. Acc. % (SplitMNIST) | Avg. Acc. % (CIFAR100) | Norm. Var. (SplitMNIST / SplitCIFAR100) |
|---|---|---|---|
| | | Particle Methods | |
| *Weighted Particle Filter* | 72.0 | 23.9 | 0.002 / **0.001** |
| Averaging Particles | 53.4 | 21.3 | 0.012 / 0.020 |
| Baseline Particle Filter | 50.1 | 19.8 | **0.001** / 0.006 |
| Gradient Descent | 48.7 | 20.1 | 0.032 / 0.001 |
| | | Continual Learning Baselines | |
| EWC | 66.3 | 23.2 | 0.186 / 0.010 |
| LWF | 67.3 | 26.4 | 0.097 / 0.050 |
| SI | 58.6 | 22.9 | 0.168 / 0.005 |
| | | Continual Learning Baselines with Weighted Particle Filter | |
| *EWC + WPF* | 76.8 | 25.8 | 0.004 / 0.004 |
| *LWF + WPF* | **79.2** | **29.0** | 0.012 / 0.007 |
| *SI + WPF* | 67.6 | 24.6 | 0.025 / 0.001 |

Table 2: Comparison of Average Normalized Return and Variance of Normalized Return for Particle Methods and LRL baselines with Supervised BC and PPO on both continual and lifelong setups of Dodgeball, Starpilot, Fruitbot. Higher return and lower variance is better. Our methods are in **bold**.

| Method | Average Normalized Return | | | Variance of Normalized Return | | |
|---|---|---|---|---|---|---|
| | Dodgeball | Starpilot | Fruitbot | Dodgeball | Starpilot | Fruitbot |
| | | | **Particle Methods CL** | | | |
| Supervised BC | 0.28 | 0.35 | 0.33 | 0.16 | 0.11 | 0.13 |
| Supervised BC + Weighted Particle Filter | **0.63** | **0.52** | **0.48** | **0.08** | **0.04** | **0.05** |
| Supervised BC + Averaging Particles | 0.31 | 0.41 | 0.36 | 0.11 | 0.08 | 0.07 |
| Supervised BC + Baseline Particle Filter | 0.37 | 0.33 | 0.39 | 0.09 | 0.10 | 0.07 |
| | | | **Particle Methods LRL** | | | |
| PPO (Gradient Descent) | 0.31 | 0.38 | 0.47 | 0.09 | 0.05 | 0.05 |
| PPO + Weighted Particle Filter | **0.40** | **0.55** | **0.63** | **0.04** | **0.03** | **0.03** |
| PPO + Averaging Particles | 0.34 | 0.40 | 0.44 | 0.09 | 0.03 | 0.04 |
| PPO + Baseline Particle Filter | 0.33 | 0.35 | 0.48 | 0.11 | 0.06 | 0.06 |
| | | | **LRL Baselines** | | | |
| PPO + TRAC | 0.69 | 0.62 | 0.76 | 0.16 | 0.16 | 0.16 |
| PPO + TRAC + Weighted Particle Filter | **0.74** | **0.68** | **0.80** | **0.04** | **0.01** | **0.04** |
| PPO + EWC | 0.37 | 0.40 | 0.60 | 0.11 | 0.02 | 0.04 |
| PPO + EWC + Weighted Particle Filter | **0.42** | **0.48** | **0.64** | **0.06** | **0.01** | **0.01** |

## 4.1 AVOIDING CATASTROPHIC FORGETTING AND LOSS OF PLASTICITY

**Performance Against Other Filters:** Tables 1 and 2 provide a summary of the performance comparison between our weighted particle filter and the baseline particle filters in both continual learning and lifelong RL experiments. Our weighted particle filter consistently achieves higher mean accuracy (averaged over classes and permutations) on SplitMNIST, SplitCIFAR100, and ProcGen Behavior Cloning datasets compared to the baseline particle filter, averaging particles, and traditional gradient descent (single particle). Furthermore, Figure 1 shows that our weighted particle filter is more resistant to loss of plasticity in LRL experiments compared to PPO using gradient descent. This underscores the advantage of incorporating particle weighting into the training process. This effect may align with the conclusions of Lyle et al. (2023); Sokar et al. (2023); Muppidi et al. (2024); Kumar et al. (2023), suggesting that when a model is not too specialized for a specific task, it is better able to adapt to new tasks. In our approach, maintaining multiple particles—some tuned to domain-specific tasks and others oriented towards different sequential tasks—enables the agent to switch to well-performing particles when adapting to new environments, thereby preserving plasticity. While theoretically, the baseline particle filter has the advantages of a Bayesian approach, because of the lack of gradient-based optimization, it fails. This lack of gradient-based optimization in high dimensions means it is essentially making random guesses, leading to performances that are close to what would be expected by chance. While gradient descent might show improved results in the latest epoch, it typically does so at the expense of previous epochs' performance. Therefore,

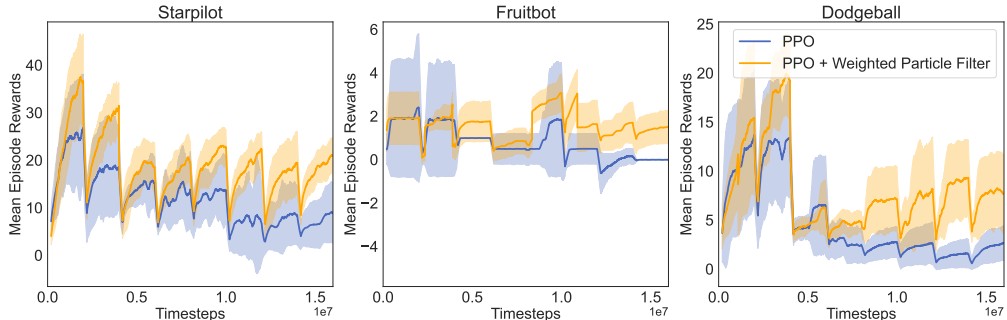

Figure 1: Mean episode reward curves for the lifelong setups of Starpilot, Fruitbot, and Dodgeball, comparing PPO and PPO with the weighted particle filter. Margins indicate standard errors over 10 runs. The results indicate that PPO with the weighted particle filter has greater resistance to the loss of plasticity observed in standard PPO.

when the performance is averaged across all epochs, the result is diminished, approaching close to random chance levels.

**Performance Compared to and Combined with Continual Learning and LRL Methods:** The results presented in Table 1 indicate that our weighted particle filter not only successfully avoids catastrophic forgetting but also outperforms all other methods in the SplitMNIST setting. In the SplitCIFAR100 dataset, our model closely competes with the top-performing continual learning model, LWF. **The greatest benefit is observed when we combine continual learning methods with our weighted particle filter. In all cases, the addition of the weighted particle filter increases accuracy.** In the lifelong RL experiments, our weighted particle filter consistently outperforms PPO + EWC across all games. Similar to the continual learning experiments, we observe that combining the weighted particle filter with PPO + EWC or PPO + TRAC results in an increase in average normalized return, demonstrating the effectiveness of these combined approaches.

**Permutation Invariance** A distinctive feature of our gradient-based weighted particle filter is its approximate permutation invariance. To validate this property, we evaluated the average normalized variances over classes or levels across 10 permutation runs for each experiment and each method in both the continual learning and lifelong reinforcement learning setups. Each run involved training on a different order of class datasets for SplitMNIST and SplitCIFAR100, or on a different order of levels for the ProcGen games. Tables 1 and 2 show that, in all experiments, our Weighted Particle Filter exhibited lower variance compared to gradient descent. Additionally, when comparing continual learning or lifelong RL methods with and without the particle filter, we observe that the Weighted Particle Filter consistently increased performance/return and reduced variance. Appendix E Figure 3 effectively illustrates this relationship in the SplitMNIST and SplitCIFAR100 experiments. The bottom right region of each plot represents the ideal scenario of high accuracy and low task variance. It is evident from both plots that this optimal region is dominated by either the Weighted Particle Filter alone or continual learning methods combined with the Weighted Particle Filter, demonstrating the advantages of our approach.

## 5 DISCUSSION

We proposed a simple, gradient-based weighted particle filter that was effective in continual, lifelong, and permutation-invariant learning. While effective, our particle filter requires memory scaling with the number of particles, which can be computationally costly. Moreover, our particle filter alone is often not as effective as in combination with other approaches, limiting its use as a standalone algorithm. Nevertheless, our approach has a number of advantages: it is highly parallelizable, requiring minimal interaction between particles. It is also domain-agnostic: our approach resists catastrophic forgetting and loss of plasticity in both lifelong reinforcement learning and supervised continual learning settings. Finally, it is easily combined with existing algorithms. These advantages suggest that particle filter methods may offer many other benefits to modern machine learning.

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

## A PROOF OF THEOREM 1

*Proof.* We first show that $\hat{p}[L_1, L_2]$ is similar to $\hat{p}[L_2, L_1]$. Observe that applying true Bayesian updates $L_1$ and $L_2$ to $\hat{p}$ following Equation 3 yields:

$$Z\hat{p}(x)e^{-L_1(x)-L_2(x)} \tag{26}$$

for some normalizing constant $Z$, which is invariant to the ordering of the loss functions. By Equation 8, we have:

$$D(\hat{p}[L_1, L_2], Z\hat{p}(\cdot)e^{-L_1(\cdot)-L_2(\cdot)}) \leq C^2 D(\hat{p}, \hat{p}) + \epsilon(C+1) = \epsilon(C+1) \tag{27}$$

$$\square$$

since $D(\hat{p}, \hat{p}) = 0$. Similarly, we have:

$$D(\hat{p}[L_2, L_1], Z\hat{p}(\cdot)e^{-L_1(\cdot)-L_2(\cdot)}) \leq \epsilon(C+1) \tag{28}$$

By the triangle inequality, we have:

$$D(\hat{p}[L_1, L_2], \hat{p}[L_2, L_1]) \leq 2\epsilon(C+1) \tag{29}$$

Now, we bound the discrepancy between $\hat{p}[L_1, L_2]$ and $\hat{p}[L_2, L_1]$ when we apply $k$ additional updates $L_3$ through $L_{2+k}$. By Equation 6, we have:

$$D(\hat{p}[L_1, L_2, L_3, ...L_{2+k}], \hat{p}[L_2, L_1, L_3, ...L_{2+k}]) \leq 2\epsilon C^k(C+1) \tag{30}$$

We may apply this inequality to bound the discrepancy between particle filter outputs when any two adjacent losses are swapped:

$$D(\hat{p}[L_1, L_2, ...L_{i-1}, L_i, L_{i+1}, L_{i+2}, ...L_T], \hat{p}[L_1, L_2, ...L_{i-1}, L_{i+1}, L_i, L_{i+2}, ...L_T]) \leq 2\epsilon C^{T-2}(C+1) \tag{31}$$

Thus, with $N$ swaps, using the triangle inequality, the discrepancy may be bounded as:

$$D(\hat{p}_0[L_1, ...L_T], \hat{p}_0[L_{\sigma_1}, ...L_{\sigma_T}]) \leq 2N\epsilon C^{T-2}(C+1) \tag{32}$$

## B    PROOF OF THEOREM 2

*Proof.* We first bound the difference in loss between $\hat{p}_0[L_1, L_2, ...L_{i-1}, L_i, L_{i+1}, L_{i+2}, ...L_T]$ and $\hat{p}_0[L_1, L_2, ...L_{i-1}, L_{i+1}, L_{i+2}, ...L_T, L_i]$. By Theorem 1, we have:

$$D(\hat{p}_0[L_1, L_2, ...L_{i-1}, L_i, L_{i+1}, L_{i+2}, ...L_T], \hat{p}_0[L_1, L_2, ...L_{i-1}, L_{i+1}, L_{i+2}, ...L_T, L_i]) \leq 2T\epsilon C^{T-2}(C+1) \tag{33}$$

Applying the bound on the difference of $L$ under different distributions:

$$\mathbb{E}_{x \sim \hat{p}_0[L_1, L_2, ...L_{i-1}, L_i, L_{i+1}, L_{i+2}, ...L_T]}[L_i(x)] - \mathbb{E}_{x \sim \hat{p}_0[L_1, L_2, ...L_{i-1}, L_{i+1}, L_{i+2}, ...L_T, L_i]}[L_i(x)] \leq 2kT\epsilon C^{T-2}(C+1) \tag{34}$$

Now, applying the reduction in loss by training on $L_i$:

$$\mathbb{E}_{x \sim \hat{p}_0[L_1, L_2, ...L_{i-1}, L_i, L_{i+1}, L_{i+2}, ...L_T]}[L_i(x)] \leq \beta \mathbb{E}_{x \sim \hat{p}_0[L_1, L_2, ...L_{i-1}, L_{i+1}, L_{i+2}, ...L_T]}[L_i(x)] + 2kT\epsilon C^{T-2}(C+1) \tag{35}$$

$\square$

## C    PROOF OF THEOREM 3

*Proof.* First, observe that since the losses are linear, particle $i$ at time $t$ has position:

$$x_t^{(i)} = x_0^{(i)} - \sigma^2 \sum_{\tau=1}^{t} g_\tau \tag{36}$$

Next, observe that the weight of particle $i$ at the end of training may simply be expressed as the product of all weight updates:

$$w_T^{(i)} = e^{-\frac{1}{2} \sum_{t=1}^{T} L_t(x_t^{(i)}) + L_t(x_{t-1}^{(i)})} \tag{37}$$

We omit the normalizing constant for notational convenience. Using the linearity of $L_t$ and the update equation $x_t^{(i)} = x_{t-1}^{(i)} - \sigma^2 g_t$:

$$w_T^{(i)} = e^{-\sum_{t=1}^{T} g_t^T(x_{t-1}^{(i)} - \frac{1}{2}\sigma^2 g_t) + b_t} \tag{38}$$

Now, expanding $x_{t-1}^{(i)}$ in terms of $g_t$:

$$w_T^{(i)} = e^{-\sum_{t=1}^{T} g_t^T(x_0^{(i)} - \sigma^2 \sum_{\tau=1}^{t-1} g_\tau - \frac{1}{2}\sigma^2 g_t) + b_t} \tag{39}$$

Rearranging terms:

$$w_T^{(i)} = e^{-\sum_{t=1}^{T} g_t^T x_0^{(i)} + \sigma^2 \sum_{t=1}^{T}(\sum_{\tau=1}^{t-1} g_t^T g_\tau + \frac{1}{2} g_t^T g_t) - \sum_{t=1}^{T} b_t} \tag{40}$$

Rewriting the double summation:

$$w_T^{(i)} = e^{-\sum_{t=1}^{T} g_t^T x_0^{(i)} + \frac{1}{2}\sigma^2 \sum_{t=1}^{T} \sum_{\tau=1}^{T} g_t^T g_\tau - \sum_{t=1}^{T} b_t} \tag{41}$$

Rearranging terms again:

$$w_T^{(i)} = e^{-\sum_{t=1}^{T} g_t^T [x_0^{(i)} - \frac{1}{2}\sigma^2 \sum_{\tau=1}^{T} g_\tau] - b_t} \tag{42}$$

Observe that $x_T^{(i)} = x_0^{(i)} - \frac{1}{2}\sigma^2 \sum_{\tau=1}^{T} g_\tau$; thus, $x_0^{(i)} - \frac{1}{2}\sigma^2 \sum_{\tau=1}^{T} g_\tau = \frac{1}{2}(x_T^{(i)} + x_0^{(i)})$. Using this and the linearity of $L_t$:

$$w_T^{(i)} = e^{-\frac{1}{2} \sum_{t=1}^{T} L_t(x_0^{(i)}) + L_t(x_T^{(i)})} = e^{-\sum_{t=1}^{T} L_t(x_T^{(i)}) - \sigma^2 \frac{1}{2} \sum_{t=1}^{T} L_t(\sum_{\tau=1}^{T} g_\tau)} \tag{43}$$

Next, note that $p_T(x_T^{(i)})$ is given by:

$$p_T(x_T^{(i)}) = p_0(x_T^{(i)}) e^{-\sum_{t=1}^{T} L_t(x_T^{(i)})} \tag{44}$$

where we again omit normalizing constants for convenience. Finally, applying the same equations for particle $j$

$$\frac{w_T^{(i)}}{w_T^{(j)}} = \frac{e^{-\sum_{t=1}^{T} L_t(x_T^{(i)}) - \sigma^2 \frac{1}{2} \sum_{t=1}^{T} L_t(\sum_{\tau=1}^{T} g_\tau)}}{e^{-\sum_{t=1}^{T} L_t(x_T^{(i)}) - \sigma^2 \frac{1}{2} \sum_{t=1}^{T} L_t(\sum_{\tau=1}^{T} g_\tau)}} = \frac{e^{-\sum_{t=1}^{T} L_t(x_T^{(i)})}}{e^{-\sum_{t=1}^{T} L_t(x_T^{(j)})}} = \frac{p_T(x_T^{(i)})}{p_T(x_T^{(j)})} \tag{45}$$

$\square$

## D    EXPERIMENTAL SETUP AND DETAILS

**SplitMNIST task:**    In this task, our objective is to sequentially address a series of five binary classification tasks derived from the MNIST dataset (Creative Commons license). These tasks are designed to distinguish between pairs of digits, presenting a unique challenge in each case. The specific pairings are as follows:

- Digits 0 and 1 ({0v1})
- Digits 2 and 3 ({2v3})
- Digits 4 and 5 ({4v5})
- Digits 6 and 7 ({6v7})
- Digits 8 and 9 ({8v9})

**Split CIFAR100 Task:**    This task involves the sequential solution of 20 different 5-class classification tasks. Each task is associated with a distinct category comprising a specific group of objects or entities. The categories, along with their corresponding class labels, are listed below:

- Aquatic mammals: {beaver, dolphin, otter, seal, whale}
- Fish: {aquarium fish, flatfish, ray, shark, trout}
- Flowers: {orchid, poppy, rose, sunflower, tulip}
- Food containers: {bottle, bowl, can, cup, plate}
- Fruit and vegetables: {apple, mushroom, orange, pear, sweet pepper}
- Household electrical devices: {clock, computer keyboard, lamp, telephone, television}
- Household furniture: {bed, chair, couch, table, wardrobe}
- Insects: {bee, beetle, butterfly, caterpillar, cockroach}
- Large carnivores: {bear, leopard, lion, tiger, wolf}
- Large man-made outdoor things: {bridge, castle, house, road, skyscraper}
- Large natural outdoor scenes: {cloud, forest, mountain, plain, sea}
- Large omnivores and herbivores: {camel, cattle, chimpanzee, elephant, kangaroo}
- Medium-sized mammals: {fox, porcupine, possum, raccoon, skunk}
- Non-insect invertebrates: {crab, lobster, snail, spider, worm}
- People: {baby, boy, girl, man, woman}
- Reptiles: {crocodile, dinosaur, lizard, snake, turtle}
- Small mammals: {hamster, mouse, rabbit, shrew, squirrel}
- Trees: {maple tree, oak tree, palm tree, pine tree, willow tree}
- Vehicles 1: {bicycle, bus, motorcycle, pickup truck, train}
- Vehicles 2: {lawn mower, rocket, streetcar, tank, tractor}

**ProcGen Environment:**   We use the ProcGen (MIT license) games Starpilot, Dodgeball, and Fruitbot, which employ procedural content generation to create new levels (corresponding to specific seeds) upon episode reset. We specifically use the hard mode to introduce distribution shifts and ensure the tasks are sufficiently challenging for both lifelong reinforcement learning and continual behavioral cloning.

**Level Characteristics:**

- **Observation**: The observation space consists of an RGB image of shape 64x64x3, representing the state of the environment at each time step.
- **Action Space**: The action space is discrete, with up to 15 possible actions depending on the game.
- **Reward**: Rewards are provided in either dense or sparse formats, depending on the specific game.
- **Termination Condition**: A boolean value indicates whether the episode has ended.

**Offline Data Collection:**

- **Levels Used**:
  - Levels **[0, 15)**: These levels are used for collecting trajectories, specifically for the lifelong RL setup. The agent in our BC experiments is trained sequentially, observing level 0 for 2 million steps, followed by level 1, and so on.
- **Expert Policies Training**:
  - We follow a similar setup to Mediratta et al. (2024). A strong and well-trained PPO policy is used, which was trained for 20 million steps on 200 levels of each game. This approach ensures that the policy generalizes well and acts as a proficient expert agent for collecting trajectories.
- **Dataset Generation**:
  - **Expert Dataset**: To generate the expert dataset, we rolled out the final checkpoint of the pretrained PPO model (i.e., the expert policy) across the 15 training levels (splits), collecting 100,000 transitions per level.

**Lifelong RL Setup:**   For the lifelong reinforcement learning setup, we followed the same experimental protocol as Muppidi et al. (2024). In the ProcGen experiments, individual game levels were generated using a seed value as the *start_level* parameter, which was incremented sequentially to create new levels. Every 2 million steps, a new level was introduced to the agent using the hard distribution mode. To assess permutation invariance, the sequence of start-level seeds was permuted 10 times, providing a diverse set of training orders for evaluation.

**Model details:**   Our Gradient-based particle filter uses 100 particles. Particles are initialized randomly from PyTorch's nn.module network parameters. A small amount of noise is injected into these parameters in the beginning of training to increase exploration of the solution space. Our Gradient-descent implementation uses the same code, except we initialize the particle filter with only one particle. The averaging particle filter simply takes the average of the accuracies of all of the particles. The baseline particle filter does the following:

1. Resamples particles from the existing pool with probabilities proportional to the exponential of the negative loss associated with each particle. .
2. Applies perturbations to particles, enabling the exploration of the solution space.
3. Updates the weights of the particles based on the new loss.

We have also incorporated three continual learning methods: SI, EWC, and LWF (van de Ven et al., 2022). Each of these methods has been implemented following the default, method-specific settings as prescribed in the (van de Ven et al., 2022) code implementation. These three models used a "pure-domain" setting.

**For a detailed implementation of our particle filter, please refer to our code submission.** Experiments are run on a computing cluster with GPUs ranging in memory size from 11 GB to 80 GB.

# E    ADDITIONAL RESULTS

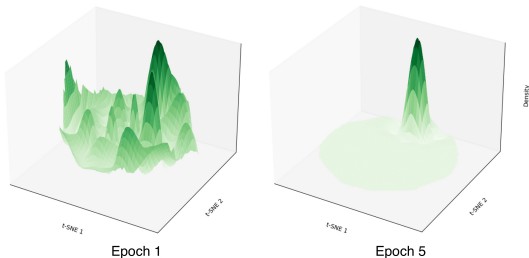

Epoch 1                    Epoch 5

Figure 2: Illustration of how our particle filter converges to well-performing regions of the parameter space over the course of training on SplitMNIST. The plot is constructed by using tSNE to map the particles into two dimensions, then representing each particle with a unimodal Gaussian of fixed variance.

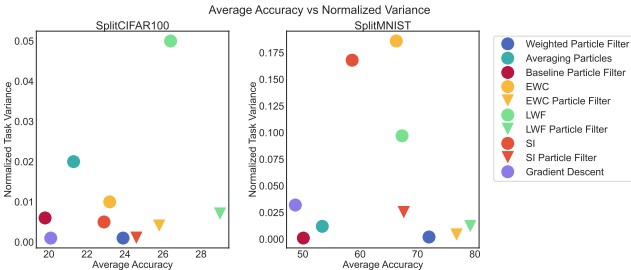

Figure 3: Average accuracy versus normalized task variance plots for both SplitCIFAR100 and SplitMNIST. The bottom right region of each plot represents the ideal scenario of high accuracy and low task-specific variance.

