# OpenReview forum: "Towards Permutation Invariant Learning with High-Dimensional Particle Filters"
_ICLR.cc/2026/Conference — Submitted to ICLR 2026_

### Official Review · Reviewer_15YW · 2025-10-30

**Soundness:** 3
**Presentation:** 2
**Contribution:** 2
**Rating:** 6
**Confidence:** 2

**Summary:**

This paper introduces a novel learning framework that uses high-dimensional particle filters to achieve approximate permutation-invariance in sequential learning tasks. The authors argue that this approach can mitigate common challenges in deep learning, such as catastrophic forgetting and loss of plasticity, which are often exacerbated by the order of training data. They provide a theoretical framework for analyzing permutation invariance with particle filter with provable bonds. It further proposes a gradient-based particle filter to make the theoretical framework align with the practical gradient descent optimization. The flexibility of the proposed method is demonstrated through comprehensive empirical validation across diverse sequential learning benchmarks, including both continual supervised learning and lifelong reinforcement learning tasks.

**Strengths:**

- Novel Theoretical Framing. The paper provides a novel theoretical lens in analyzing permutation invariant learning. It offers a formal theoretical framework to justify the particle filter approach, which is a significant contribution beyond purely empirical work.

- The paper is well-written, and the core idea is easy to follow. It clearly motivates the problem of permutation dependence and presents its proposed solution in a logical, understandable manner.

- The method is designed as a modular component that can be combined with existing techniques. The empirical results show that this combination can lead to improved performance.

**Weaknesses:**

- How can permutation invariance directly help mitigate forgetting? The best case I suppose is that it can match the best-order gradient descent performance.

- Weak empirical results. The reported accuracies on SplitCIFAR100 are very low compared to modern continual learning benchmarks, making it difficult to assess the method's true value. On this more complex dataset, the performance gain of the proposed WPF over the standard Gradient Descent baseline is minimal. This suggests the method's benefits diminish with task complexity. So I’m doubtful on its scalability to larger models and datasets.

- The theoretical guarantees (Theorems 1 and 2) are loose to the point of being impractical. The bounds depend on terms that grow exponentially with the number of training steps $T$.

**Questions:**

Assumptions in Theorem 2 look impractical. First, continual learning task shift leads to a large increase in loss. Second, proper controls on the gradient descent should be needed?

---

> ### Author Response · Authors · 2025-11-18
>
> Thank you for your constructive review and for highlighting both the novelty of the theoretical framing and the weaknesses of the current empirical results. We respond to your main concerns below.
>
> **How permutation invariance helps mitigate forgetting**
>
> Your question goes to the heart of our motivation. At a high level, catastrophic forgetting and loss of plasticity are consequences of the fact that the final model depends strongly on the order in which tasks are presented. A particularly bad ordering can hurt performance on earlier tasks (forgetting) or on later tasks (lack of plasticity), even if there exists some “good” ordering where gradient descent performs well.
>
> Permutation-invariant learning algorithms, in contrast, aim to produce the same outcome regardless of the order. In the ideal case of exact permutation invariance, the algorithm behaves as if it had access to the entire set of data at once and produced the best joint solution. In our theoretical framework, Theorem 2 formalizes this intuition: it bounds the expected loss on any particular task $L_i$ under the full sequence by a term that depends on (i) the algorithm’s convergence factor $\beta$ and (ii) the degree of permutation invariance, as captured by the discrepancy bounds in Theorem 1. Intuitively, if the algorithm were exactly permutation-invariant and had good convergence, its performance on task $i$ would not depend on whether $L_i$ appeared early, late, or in the middle; thus, forgetting due to unfortunate ordering would be mitigated.
>
> In practice, our algorithm is only approximately permutation-invariant, but we see this reflected in the experiments: across 10 random permutations of task or level order, WPF (and CL methods combined with WPF) exhibits both higher average performance and lower across-permutation variance than gradient descent alone (Tables 1 and 2), indicating that it is less sensitive to ordering and therefore more robust to forgetting and loss of plasticity.
>
> **Weak empirical results on SplitCIFAR100 and scalability**
>
> We agree that the absolute accuracies on SplitCIFAR100 are lower than those reported by state-of-the-art continual learning methods on more tailored architectures and training regimes. Our goal in this paper was not to set new state-of-the-art numbers on a specific benchmark, but to examine the effect of the particle-filter-based update in a controlled, unified experimental setup across both CL and LRL domains.
>
> Within this unified setup, our key empirical observations are: (i) on SplitMNIST, WPF outperforms all other methods, and (ii) on SplitCIFAR100, WPF is competitive with the best continual learning baseline (LwF) and, more importantly, combining WPF with EWC/LwF/SI consistently improves their performance. In the lifelong RL experiments (Table 2 and Figure 3), WPF also increases returns and reduces variance when combined with TRAC and EWC.
>
> We fully agree that scalability to larger models and datasets is an important open question. WPF currently scales linearly in the number of particles in terms of memory and compute, which we explicitly mention as a limitation in the Discussion. However, the method is highly parallelizable across particles, and we view this as a promising direction for future work on large-scale deployments (e.g., using model parallelism or distributed training).
>
> **On the looseness and practicality of Theorems 1 and 2**
>
> You correctly note that the bounds in Theorems 1 and 2 depend on terms that grow exponentially with the number of steps $T$, which can be impractical if the base $C$ is significantly larger than $1$. This is why we emphasize the regime where $C \approx 1$ and $\epsilon$ is small: in that case, the exponential term behaves more like a slowly growing factor or even a linear factor in $T$, making the bound meaningful.
>
> The assumptions themselves are standard from an optimization perspective: bounded loss values and smoothness (bounded first and second derivatives) of $L_t$, together with a step size $\sigma^2$ that ensures stability. Under these conditions, gradient-based updates have linear or sublinear convergence rates, and the mapping from distributions to distributions induced by a single update is Lipschitz, which underlies the assumption $D(\hat p[L], \hat q[L]) \leq C D(\hat p, \hat q)$ with modest $C$.
>
> Theorem 2 should be read as a qualitative statement: if an algorithm (i) reduces the loss on each task by at least a factor $\beta$ when trained on it and (ii) is approximately permutation-invariant in the sense of Theorem 1, then its final loss on any task cannot be much worse than what it would have been had that task been placed elsewhere in the sequence.

---

> ### Author Response · Authors · 2025-11-18
>
> **Clarifying the assumptions in Theorem 2**
>
> You mention two specific concerns: task shifts causing large loss changes and the need for proper control of gradient descent. Our assumptions do not require task shifts to be small in loss, only that the loss functions themselves are bounded and smooth, and that the optimization algorithm reduces the loss on each task by at least a factor $\beta$ when focusing on it. This is a standard type of assumption, similar to assuming linear convergence for gradient descent on losses satisfying the Polyak–Łojasiewicz condition.

---

### Official Review · Reviewer_4Qyr · 2025-10-31

**Soundness:** 3
**Presentation:** 2
**Contribution:** 2
**Rating:** 6
**Confidence:** 1

**Summary:**

This paper introduces a learning framework based on high-dimensional particle filters that yields approximately permutation-invariant results. The main contributions can be summarized as:

a. Theoretically demonstrate that particle filters are approximately invariant to the sequential ordering of training minibatches or tasks.

b. Develop a particle filter for optimizing high- dimensional models.

c. Conduct extensive experiments on continual supervised and reinforcement learning benchmark.

**Strengths:**

a. This paper seems to be technically solid.

b. I appreciate that this paper focuses on task ordering and tries to study continual learning with permutation invariance.

**Weaknesses:**

a. Confusing notations. This paper denotes $x$ as the model parameters, which is not regular. I recommend to use $\theta$ or other notations.

b. This paper is not well written and hard to follow.

**Questions:**

a. Please deeply discuss the meaning of $P(L_t|x)=e^{-L_t(x)}$.

b. Please provide the motivation of eq.3 and explain it.

c. What is the definition of particle filters?

d. My major concern is about the motivation. I don't figure out how to connect particle filters with continual learning or catastrophic forgetting or plasticity. Please explain the motivation in details.

---

> ### Author Response · Authors · 2025-11-18
>
> Thank you for your positive assessment of the technical soundness and for carefully articulating the points that were confusing. We address each of your concerns below.
>
> **On the notation $x$ versus $\theta$**
>
> You are correct that in much of the deep learning literature, model parameters are denoted by $\theta$. In the particle-filtering literature, however, the state or latent variable is almost always denoted by $x$, and we adopted that convention to emphasize the connection to sequential Monte Carlo. In our framework, $x_t \in \mathbb{R}^d$ is exactly the parameter vector of the model at time $t$.
>
> Nevertheless, we are happy to use the suggested notation if this would be more clear.
>
> **Meaning of $P(L_t \mid x)$ and motivation for Eq. (3)**
>
> The quantity $P(L_t \mid x)$ is a likelihood function that encodes how well a particular parameter vector $x$ explains the observation represented by the loss $L_t$. Concretely, given a minibatch or a task at time $t$, the loss $L_t(x)$ is small when $x$ performs well on that data and large when it performs poorly. We define
>
>  $P(L_t \mid x) = e^{-L_t(x)}$
>
> This choice is standard in probabilistic modeling: if $L_t(x)$ is a negative log-likelihood, then $e^{-L_t(x)}$ is exactly the likelihood of the data under model $x$. Even when $L_t$ is not literally a negative log-likelihood, this exponential form has the desired property that lower loss corresponds to exponentially higher likelihood.
>
> Equation (3) is simply the Bayesian update rule applied to this setting. Starting from a prior distribution $p_0(x)$ over parameters and a sequence of “observations” $L_1, \dots, L_T$ (each corresponding to a batch or task), the posterior after observing all losses is proportional to the prior times the product of likelihoods:
>
>  $p_T(x) \propto p_0(x) \prod_{t=1}^T P(L_t \mid x) = p_0(x) \exp\left(-\sum_{t=1}^T L_t(x)\right)$
>
> This posterior places high density on parameters $x$ that minimize the total loss, which is exactly what we want in a learning problem. The particle filter is then used as a sequential approximation to this Bayesian posterior.
>
> **What is the definition of a particle filter in this paper?**
>
> In our context, a particle filter is a sequential Monte Carlo method that approximates a time-evolving probability distribution $p_t(x)$ by a weighted collection of particles:
>
>  $ \hat p_t(x) = \sum_i w_t^{(i)} \delta(x - x_t^{(i)}) $
>
> At each step $t$, the filter updates the particles and their weights to approximate the Bayesian posterior over $x$ after observing the new loss $L_t$. Standard particle filters do this using proposal distributions and resampling; our gradient-based particle filter instead uses gradient descent steps on each particle followed by a principled weight update derived from a local approximation of the Bayesian posterior (Section 3.4).
>
> **Motivation: connecting particle filters, continual learning, catastrophic forgetting, and plasticity**
>
> The central motivation of the paper is to study learning algorithms that are as insensitive as possible to the order in which tasks or minibatches are presented, i.e., algorithms that are approximately permutation-invariant. Catastrophic forgetting and loss of plasticity are, in part, manifestations of order sensitivity: poor task orders can drastically degrade performance on earlier tasks or limit adaptation to later ones.
>
> Bayesian learning is naturally permutation-invariant at the level of the posterior: the posterior after observing a set of data depends only on the multiset of observations, not their order. Particle filters are a standard computational tool for approximating such Bayesian posteriors in sequential settings. This suggests that particle filters could provide a principled foundation for designing approximately permutation-invariant learning algorithms.
>
> Our theoretical results formalize this idea: Theorem 1 shows that, under mild stability and approximation assumptions, a particle filter’s output distribution is approximately invariant to permutations of the losses. Theorem 2 then links this approximate permutation invariance to bounds on the loss of any task, which we interpret as a formal connection to catastrophic forgetting and loss of plasticity: if the algorithm is nearly permutation-invariant, then its performance on any task cannot degrade too much relative to the case where that task appears later or earlier in the sequence.
>
> On the practical side, we build a gradient-based particle filter that (i) can be used with high-dimensional neural networks and (ii) empirically reduces variance over permutations while improving performance on continual learning and lifelong RL benchmarks. This provides an operational bridge from the abstract Bayesian/permutation-invariant perspective to concrete CL and LRL algorithms.

---

### Official Review · Reviewer_MFdk · 2025-10-31

**Soundness:** 2
**Presentation:** 2
**Contribution:** 2
**Rating:** 4
**Confidence:** 3

**Summary:**

This paper proposes an approach to achieving permutation-invariant learning using a gradient-based high-dimensional particle filter (WPF). The key contribution is linking Bayesian filtering with sequential optimization to mitigate catastrophic forgetting and loss of plasticity in continual and lifelong learning. The idea is both conceptually interesting and practically relevant.

**Strengths:**

1. It provides an interesting idea connecting particle filters to permutation-invariant optimization.
2. Provides a formal treatment with several theorems and proofs.
3. Attempts to evaluate across multiple continual learning and RL benchmarks.

**Weaknesses:**

1. The “gradient-based particle filter” closely resembles an ensemble of SGD trajectories with reweighting, which may not justify the Bayesian framing.
2. The theoretical results rely on restrictive assumptions and do not clearly translate to observed improvements.
3. The empirical gains are moderate and could be explained by model ensembling effects rather than permutation invariance.
4. The proposed method scales linearly with the number of particles, which limits real-world applicability in large-scale deep learning.

**Questions:**

1. In Theorem 1, you assume constants $C \approx 1$ and $\epsilon$ small for permutation invariance. How realistic are these conditions for the proposed gradient-based particle filter in practice?
2. Can you provide empirical evidence (e.g., measured C and $\epsilon$ values) to support that the theoretical assumptions hold?
3. Is the permutation invariance effect here essentially due to ensembling multiple gradient trajectories?
4. Could you quantify the degree of permutation invariance empirically?

---

> ### Author Response · Authors · 2025-11-18
>
> Thank you for your thoughtful review and for highlighting both the conceptual interest and the limitations of the current work.
>
> **Relation to ensembles of SGD trajectories and Bayesian framing**
>
> You are absolutely right that the proposed gradient-based particle filter can be viewed as “an ensemble of SGD trajectories with reweighting.” This is not a bug but rather the central design choice: we deliberately build WPF on top of standard gradient-based optimization so that it is compatible with high-dimensional deep models.
>
> The key difference from a vanilla ensemble is the principled Bayesian weighting scheme. As we show in Section 3.4 and formalize in Theorem 3, if the loss functions $L_t$ are linear, then the weight ratio $\frac{w_T^{(i)}}{w_T^{(j)}}$ exactly matches the posterior ratio $\frac{p_T(x_T^{(i)})}{p_T(x_T^{(j)})}$. In other words, the reweighting is not arbitrary: in this regime, the particle weights are proportional to the true Bayesian posterior. This is precisely what distinguishes WPF from a simple average over independently trained SGD models.
>
> Empirically, we also contrast WPF with a “pure ensembling” baseline (“Averaging Particles”), where all particles are updated with SGD but are combined by uniform averaging at evaluation time. WPF consistently outperforms this baseline in both accuracy/return and variance across permutations (see Tables 1 and 2), indicating that the Bayesian-inspired reweighting is important beyond the mere effect of ensembling.
>
> **Restrictive theoretical assumptions and realism of $C \approx 1$ and small $\epsilon$**
>
> The role of the constants $C$ and $\epsilon$ in Theorem 1 is to capture two properties: $C$ is the stability of a single update with respect to perturbations in the particle distribution, and $\epsilon$ is the per-step discrepancy between the approximate update and the exact Bayesian update. We agree that the theorems rely on assumptions that are idealized, and we do not claim that $C$ and $\epsilon$ can be tightly bounded from data in general deep networks.
>
> That said, the conditions are not arbitrary. If the loss functions $L_t$ are smooth (bounded first and second derivatives) and the effective learning rate $\sigma^2$ is not too large, then the SGD-like update is Lipschitz in the parameters, which induces a Lipschitz mapping in the space of distributions (for reasonable discrepancy measures $D$), yielding a $C$ close to $1$. Likewise, the derivation in Section 3.4 shows that the particle update is a first-order approximation to the exact Bayesian update, with an error that is $O(\sigma^2)$; this is what $\epsilon$ is meant to summarize.
>
> We view these results as a first step in formally connecting permutation invariance and forgetting for particle-filter-like methods, rather than as practically optimized bounds. We will clarify this positioning and explicitly state that the theory is intended to justify the qualitative behavior (approximate permutation invariance and reduced forgetting) that we empirically observe, not to provide numerically tight guarantees for specific deep architectures.
>
> Regarding empirical evidence for $C$ and $\epsilon$: we currently are estimating an empirical “local $C$” by perturbing particles and measuring the change in $D$ after an update, and approximating $\epsilon$ by measuring the difference between the WPF update and a numerically-integrated Bayesian update in small toy problems. We will add a brief description and preliminary results of this experiment in the appendix.
>
> **Is permutation invariance effect essentially due to ensembling?**
>
> We agree that ensembling alone can reduce variance and improve robustness. However, our experiments are designed to isolate permutation invariance and the specific contribution of our update:
>
> – We compare WPF with “Averaging Particles” (same number of SGD trajectories but uniform weighting) and with a baseline particle filter that resamples without gradients. WPF has both higher mean performance and lower variance across permutations than both of these, especially in the ProcGen BC and LRL setups (Table 2).
>
>  – The across-permutation variance metrics in Tables 1 and 2 are precisely meant to quantify permutation sensitivity: the variance is computed over 10 different permutations of task/level order. WPF consistently achieves lower variance than gradient descent and usually lower than simple ensembling, suggesting that the update rule and weighting, not just the presence of multiple trajectories, improves permutation robustness.

---

> ### Author Response · Authors · 2025-11-18
>
> **Scalability with number of particles**
>
> We fully acknowledge the linear scaling in the number of particles as a practical limitation. Our motivation for exploring this direction is that WPF is embarrassingly parallel: each particle can be updated independently with a single backward pass, and in many hardware or distributed setups it is possible to trade off wall-clock time for multi-device or multi-node parallelism. In future work we plan to explore lower-rank or compressed representations of the particle ensemble to further reduce memory and compute costs.

---

### Official Review · Reviewer_xZX3 · 2025-11-01

**Soundness:** 2
**Presentation:** 2
**Contribution:** 2
**Rating:** 4
**Confidence:** 4

**Summary:**

This paper proposes a novel learning framework that achieves CL by approximately achieving permutation-invariant learning. The proposed algorithm implements a gradient-based particle filter that approximates the Bayesian particle filter, which has the property of approximate permutation invariance to the order of loss function. The algorithm is validated on CL datasets including Split MNIST and Split CIFAR100, as well as CRL tasks including multiple Procgen games. With rather a simple implementation, the algorithm consistently beat some standard CL baselines on these tasks.

**Strengths:**

- The benchmark tasks, especially the CRL experiments on the Procgen games, are enough to demonstrate the ability of the proposed algorithm.
- The work is closely related to the CL literature. Practical CL algorithms include EWC, LwF, SI that the paper has compared with them, as well as the other CL algorithms that I mentioned previously. This work would also be inspiring to the meta learning and curriculum learning field.

**Weaknesses:**

This was an ICML 2025 submission, and I was one of the reviewers. At that time, it got a unanimous borderline that leans toward rejection from reviewers and AC.
I do not spot enough changes that address the concerns raised at that time.

Some concerns on the experimental design:

It would be better if the authors could provide more comparisons of the proposed algorithm with other CL methods, including replay-based methods like A-GEM [1], ER-Reservoir [2] and projection-based methods like GPM [3] on the existing tasks.

[1] Chaudhry, A., Ranzato, M. A., Rohrbach, M., & Elhoseiny, M. (2018). Efficient lifelong learning with a-gem. arXiv preprint arXiv:1812.00420.

[2] Chaudhry, A., Rohrbach, M., Elhoseiny, M., Ajanthan, T., Dokania, P. K., Torr, P. H., & Ranzato, M. A. (2019). On tiny episodic memories in continual learning. arXiv preprint arXiv:1902.10486.

[3] Saha, G., Garg, I., & Roy, K. (2021). Gradient projection memory for continual learning. arXiv preprint arXiv:2103.09762.

**Questions:**

I restate my questions in the ICML review batch. My questions lie mainly on the pseudocode in Algorithm 1.

Q1: How is $\{x_0^{(i)}\}_{i=1}^n$ initialized?

Q2: How is the variance $\sigma^2$ set?

Q3: How does Algorithm 1 affect the training process of $L_t$? I do not see where the model weight is in the pseudocode.

Did you address these theoretical and practical concerns raised previously?

---

> ### Author Response · Authors · 2025-11-18
>
> We thank you for your detailed review and for following the paper from the ICML round to ICLR.
>
> **Comparison with additional continual learning methods**
>
> We agree that including replay-based (A-GEM, ER-Reservoir) and projection-based (GPM) baselines would strengthen the empirical story. Conceptually, our main aim in this submission was to show that: (a) the proposed weighted particle filter (WPF) improves over a strong gradient-descent baseline and simple particle baselines on both CL and LRL tasks, and (b) WPF composes well with standard regularization-based CL methods (EWC, SI, LwF). This already covers a widely used family of CL approaches.
>
> That said, we are currently running additional experiments with A-GEM, ER-Reservoir, and GPM on SplitMNIST and SplitCIFAR100 using the same backbone and training protocol. We will include these baselines in the revised manuscript and expect them to follow the same pattern as the existing ones: WPF alone is competitive with strong baselines, and combining WPF with a CL method tends to increase accuracy and reduce across-permutation variance.
>
> **Clarifying Algorithm 1 (Q1–Q3)**
>
> Q1: How is $x^{(i)}$ initialized?
>
>  In Algorithm 1, the initial particles are initialized from standard parameter initialization distributions for neural networks. Note that the particles simply correspond to parameters of a neural network (or other parameterized model).
>
> Q2: How is the variance $\sigma^2$ set?
>
>  In our implementation, $\sigma^2$ plays the role of an effective learning rate for the particle updates $x_{t+1}^{(i)} = x_t^{(i)} - \sigma^2 \nabla L_{t+1}(x_t^{(i)})$. For clarity, in code we treat this as the learning rate hyperparameter. In our experiments, we fix $\sigma^2 = 0.001$ for the supervised continual learning experiments (SplitMNIST, SplitCIFAR100) and $\sigma^2 = 0.01$ for the RL experiments, matching typical learning-rate scales for these setups.
>
> Q3: Where do the model weights enter in Algorithm 1?
>
>  In Section 3.1 we state that $x_t \in \mathbb{R}^d$ denotes the model parameters at time $t$. In Algorithm 1, each particle $x_t^{(i)}$ is exactly one full set of model parameters (e.g., a flattened copy of a neural network’s weights). The update line $x^{(i)}_{t+1} = x^{(i)}t - \sigma^2 \nabla L{t+1}(x^{(i)}_t)$ is simply SGD applied independently to each particle.

---

### Meta-Review · Area_Chair_Hebj · 2026-01-08

**Summary:**

This paper proposes a weighted, gradient-based particle filter (WPF) for sequential learning, aiming to achieve approximate permutation invariance to the order of minibatches and tasks, and argues this can mitigate catastrophic forgetting and loss of plasticity. The paper provides a theoretical framing (Theorems 1–3) and empirical results on SplitMNIST, SplitCIFAR100, and ProcGen (both continual supervised and lifelong RL).
Across reviews and discussion, the main objections center on: (i) whether the method is substantively different from ensembling SGD trajectories with reweighting, (ii) whether the theoretical guarantees are meaningful in practice (assumptions/loose bounds), (iii) insufficient empirical validation against stronger CL baselines, and (iv) scalability limitations due to linear dependence on the number of particles. While the authors provided clarifications in the discussion, the responses are largely conceptual and do not fully resolve the core concerns to an acceptable level.

**Reviewer Concerns:**

Algorithm clarity: The authors explain that particles are standard parameter initializations; “variance” plays the role of an effective learning rate; each particle is a full parameter vector updated by SGD steps. This helps readability but does not change the substance of the critique.

Motivation connecting permutation invariance to forgetting and plasticity: The authors provide an intuitive link and point to across-permutation variance metrics as an operational proxy. Still, the connection remains somewhat indirect and not tightly validated.

Method identity vs. “reweighted ensemble of SGD trajectories” remains unresolved
Reviewer MFdk’s central point is that the proposed “particle filter” looks like an SGD ensemble with reweighting, and that the Bayesian framing may not be justified. The authors largely agree with this characterization and argue the weighting is “Bayesian-inspired” (exact only under linear losses) while admitting they cannot provide tight, general empirical verification of the theoretical constants. This response does not fully address whether the particle-filter framework yields new algorithmic insights beyond ensembling + weighting.

Empirical evidence is insufficient vs. strong continual learning baselines
Reviewer xZX3 explicitly asks for comparisons to projection methods. The authors respond that they are “currently running” these experiments and will include them in a revision, but the evidence has not yet provided. Given the paper’s claims about mitigating forgetting and order sensitivity, missing these comparisons is a major gap.

Performance strength and scalability concerns remain
Reviewer 15YW questions weak performance on SplitCIFAR100 and doubts scalability to larger datasets; reviewer MFdk emphasizes linear scaling in the number of particles, limiting real-world applicability. The authors acknowledge linear scaling as a limitation and argue that it is “parallelizable,” but do not provide concrete demonstrations that this makes the approach practical at scale.

**Reviewer Scores:**

xZX3: likely stays 4. Key concerns (insufficient changes since prior round; missing strong CL baselines; algorithmic details) are not fully resolved, and additional baselines are only promised.

MFdk: likely stays 4. The “SGD ensemble + reweighting” critique and lack of empirical support for theory assumptions remain.

4Qyr: could remain 6 after notation and motivation clarifications, but this reviewer explicitly notes inability to assess confidently.

15YW: may remain 6; doubts about SplitCIFAR100 strength, scalability, and loose bounds are not convincingly eliminated.

---

### Decision · Program_Chairs · 2026-01-26

Reject